# Physical Activity Change during COVID-19 Confinement

**DOI:** 10.3390/ijerph17186878

**Published:** 2020-09-21

**Authors:** Arkaitz Castañeda-Babarro, Ane Arbillaga-Etxarri, Borja Gutiérrez-Santamaría, Aitor Coca

**Affiliations:** Health, Physical Activity and Sports Science Laboratory, Department of Physical Activity and Sports, Faculty of Psychology and Education, University of Deusto, 48007 Bizkaia, Spain; ane.arbillaga@deusto.es (A.A.-E.); borjagutierrez@deusto.es (B.G.-S.); aitor.coca@deusto.es (A.C.)

**Keywords:** confinement, physical activity, sedentary behavior, Covid-19

## Abstract

**Background:** The lockdown and social distancing caused by COVID-19 may influence common health behavior. The unprecedent worldwide confinement, in which Spain has been one of the most affected—with severe rules governing confinement—may have changed physical activity (PA) and sedentary habits due to prolonged stays at home. **Purpose:** The aim of this study is to evaluate how self-reported PA and sedentary time (ST) have changed during confinement in the Spanish population. **Methods:** 3800 healthy adults (age 18–64 years) residing in Spain answered the international physical activity questionnaire short (IPAQ-S) twice between 23 March and 1 April (confinement). Data analysis was carried out taking into consideration meeting general PA recommendations before confinement, age and gender. **Results:** Self-reported PA decreased significantly during confinement in our sample. Vigorous physical activities (VPA) and walking time decreased by 16.8% (*p* < 0.001) and 58.2% (*p* < 0.001), respectively, whereas ST increased by 23.8% (*p* < 0.001). The percent of people fulfilling the 75 min/week of VPA recommendation decreased by 10.7% (*p* < 0.001) while the percent of people who reached 150 min/week of moderate activity barely changed (1.4%). The group that performed **the most** VPA before confinement showed the greatest decrease (30.5%, *p* < 0.001). Men reduced time in VPA more than women (21% vs 9%, respectively) who even increased time in moderate PA by 11% (*p* < 0.05) and **reported** less increase in ST than men (35% vs 25.3%, respectively). **Conclusion:** The Spanish adult population, especially young people, students and very active men, decreased daily self-reported PA and increased ST during COVID-19 confinement.

## 1. Introduction

On 11 March 2020, the World Health Organization (WHO) [1] declared a global pandemic caused by severe acute respiratory syndrome coronavirus (SARS-Covid-2), which has become a public health emergency of international concern. During its first phase of expansion outside China, Italy and Spain were the most affected countries reporting most cases and deaths. Thus, they were the first nations to declare a state of emergency in Europe. In Spain it was declared on 17 March and the government ordered a lockdown to restrict travel and cancel non-essential services in order to stop the spread of coronavirus disease (COVID-19) [2].

Social distancing and confinement are fundamental in tackling the spread of coronavirus. However, the ongoing lockdown across the country has no precedent and it is unknown how this may affect the general population’s health and wellbeing. In these circumstances, the sudden and stressful situation in addition to prolonged stays at home may imply a radical change in lifestyle behavior such as physical activity (PA), eating habits, alcohol consumption, mental health, quality of sleep, etc. [3,4,5,6,7].

Likewise, there is a general concern about the negative health implications of inactivity and sedentary behavior [8]. The general recommendation for considering an adult to be physically active is to attain at least 150 min of moderate or 75 min of vigorous intensity activity per week or an equivalent combination of both [9], and sedentary behavior is defined as any waking behavior practiced while lying down, reclining, sitting or standing, involving an energy expenditure ≤ 1.5 metabolic equivalents [10]. While the disease spreads around the world, healthy people are being requested to stay at home for prolonged periods of time and, as a consequence, COVID-19 has radically modified the determining factors (individual, interpersonal, environmental, regional or national policies and global) [11] of both types of behavior Thus, due to isolation and limitations in engaging in regular and common activities, fulfilling PA recommendations and reducing sedentary behavior during lockdown may pose a significant challenge, especially during the first weeks when the population has limited chances to find alternatives to ensure they remain active even at home.

Hence, although individuals were encouraged to remain physically active in their homes [12], the unprecedented confinement may give rise to two situations. (1) the active population may decrease their activity and (2) the inactive population may not be likely to increase their daily PA.

Despite the fact that lockdown has affected several countries, data are scarce about how people have changed their PA and sedentary behavioral patterns because of the specific cases of isolation in each country. Therefore, the aim of this study was to analyze self-reported PA and sedentary behavior before and during lockdown caused by COVID-19 in a Spanish healthy adult population.

## 2. Material and Methods

### 2.1. Participants

Healthy adults (age 18–64 years, age category defined by WHO) living in Spain were asked to participate in this cross-sectional study. All the subjects were informed about the objective of the study and their free participation in it, being able to leave it whenever they wanted. Ethics approval was obtained from the Deusto University Human Ethics Advisory Group, and informed consent was obtained from participants.

### 2.2. Experimental Trials

Sociodemographic data (age, height, weight, gender, whether working and/or studying) and self-reported PA data were collected by questionnaires sent between 23 March and 1 April, just 10 days after the state of emergency was declared in Spain. Two questionnaires were sent together to be answered consecutively at the same time. The first collected retrospective data about habits during a normal week before lockdown, and the second asked about the following one or two weeks. The questionnaire used was the IPAQ short version validated in Spanish [13]. The IPAQ short version asks about three specific types of activity undertaken during the previous 7 days in the four domains (leisure time, work, household activities and transport); items are structured to provide separate scores for walking, moderately intense and vigorously intense activities. The IPAQ version also contains a question about the time spent on sedentary activity.

The invitation to participate in the study was issued via social media, e-mail and mobile phone from the Physical Activity and Sports Sciences Department, University of Deusto. Moreover, a national sports store (Forumsport S.A.) and a Biomechanical Laboratory (Custom4us) also sent the questionnaire to their customers by e-mail.

### 2.3. Procedure

The primary outcome was the change in time and intensity of self-reported PA and the sedentary time prior to the confinement situation and after it. The secondary outcome was the change in the percentage number of participants who fulfilled general PA recommendations according to age, gender, working status and baseline PA levels.

A total of 4160 healthy subjects answered the questionnaire. A total of 360 were excluded due to exclusion criteria: not resident in Spain (*n* = 16), age < 18 or ≥65 (*n* = 110) and extreme scores in vigorous, moderate and walking activities (≥6 h per day, 7 days a week). The exact number of participants excluded was as follows: *n* = 81 were excluded because the sum of the total active time was >6 h per day (vigorous + moderate + walking). Within the pre-COVID data, *n* = 17, *n* = 31 and *n* = 62 were excluded because they exercised > 180 min per session at a vigorous level of intensity, >180 at moderate intensity and >21 h (3 h per day, 7 days of week) walking, respectively. Finally, within the during-COVID data, *n* = 17, *n* = 9 and *n* = 17 were excluded because they exercised >180 min per session at vigorous intensity, >180 at moderate intensity and >21 h (3 h per day, 7 days of week) walking, respectively.

### 2.4. Statistical Analyses

A univariate analysis with paired *t*-tests was used to compare the differences in primary outcomes before and during confinement. Furthermore, a Chi^2^ frequency test was used to assess secondary outcomes, which refer to the change in the percentage number of participants who fulfilled the amount of self-reported PA in the subgroups. A subgroup analysis was performed on different age groups (18–24, 25–34, 35–44, 45–54, 55–64), gender (m, f), working status (students, active workers, people that study and work, those that reported they did nothing), and self-reported PA, categorized into vigorous activity groups (0–75, 75–150, 150–225, more than 225 min/week) and moderate activity groups (0–150, 150–300, 300–450, >450 min/week). The PA subgroup categories are based on World Health Organization (WHO) recommendations. We divided the data into different groups following in accordance with WHO recommendations for activity of moderate (150 min per week) and vigorous (75 min per week) intensity. Within each of the two categories we divided the data into 4 groups as follows: (1) under the amount recommended (less than 150 min of moderate intensity or less than 75 min of vigorous intensity), (2) following recommendations (150 to 300 min of moderate intensity or 75 to 150 min of vigorous activity), (3) twice the amount recommended and (4) three times the amount recommended. This categorization was made to extrapolate the results to the amount of self-reported PA fulfilled by participants in the study, with α level set at 0.05. Statistical analysis was performed using the SPSS Data Analysis Version 23 (SPSS, Inc., Chicago, IL, USA).

## 3. Results

Table 1 shows the demographic characteristics of 3800 participants who were mostly male, (54%) active workers (78%) and aged 42.7 ± 10.4 years (mean ± SD).

Using the WHO stratification, we divided the sample into five BMI groups with *n* = 77 (2.1%) being the underweight group, *n* = 2621 (70.0%) normal; *n* = 897 (24.0%) overweight; *n* = 144 (3.8%) obese and *n* = 4 (0.1%) extremely obese (<18.5 kg/m^2^; 18.5–24.9; 25–29.9; 30–39, 9 and ≥40 groups of BMI score, respectively).

During confinement, the amount of time spent on moderate and vigorous activities by all the population decreased by 2.6% (*p* = 0.102) and 16.8% (*p* < 0.001), respectively. In addition, walking time was reduced by 58.2% (*p* < 0.001) whereas sedentary time increased by 23.8% (*p* < 0.001) (Table 2).

Men reported a higher decrease in vigorous activities than women (21% and 9%, respectively) and both reduced walking time to a similar extent (58.5% men, 59.6% women) (Table 2). However, sedentary time was reported to have a higher increase in men (35%, *p* < 0.001) than in women (25.3%, *p* < 0.001) and accordingly, men significantly reduced moderate activities by 8.2% (*p* < 0.001) while women increased these activities by 11% (*p* < 0.05).

The student group showed the highest decrease in moderate (16.1% *p* < 0.05), vigorous (24.3%, *p* < 0.001) and walking activities (66.9% *p* < 0.001), whereas unemployed or non-students were proved to be the most sedentary during confinement (47.7%, *p* < 0.001) (Table 2). The adult population (age 55–65 years) decreased the amount of time they spent on vigorous activities the most (22.1%) whereas for moderate activities and walking time it was the youngest subjects (18–24 years) who also evidenced the greatest increase in sedentary time (47.7% *p* < 0.001).

Regarding meeting PA recommendations, the number of subjects who failed to complete 75 minutes’ activity of vigorous intensity per week before confinement increased by 10.7% (*p* < 0.001) during lockdown. (Table 3). In addition, the most active population (>225 min/week of vigorous activity) decreased their activity significantly by 7.7% (*p* < 0.001). Nevertheless, meeting 150 min of moderate activity per week barely changed (1.4%, *p* value 0.117).

In the subgroup analysis, the most active subjects showed the highest decrease in vigorous activity time (30.5%, *p* < 0.001) (Table 2). However, the less active population increased the time they spent on these activities by 34% and this trend is also evidenced in the case of moderate activities. Furthermore, walking time declined similarly for all self-reported PA levels, while in terms of sedentary time, the most active group reported the highest increase (40.3%, *p* < 0.001).

## 4. Discussion

Self-reported PA decreased significantly during confinement in all the population, in which vigorous and walking activities declined the most and moderate activities barely changed. There were more inactive people who failed to fulfil the 75 min/week of vigorous activities during lockdown and sedentary time also increased considerably. The impact on active and sedentary behavior was particularly high in men, young people, students and the physically very active population.

To our knowledge, there are limited previous studies that have analyzed the effect of confinement caused by a pandemic on self-reported PA. Thus, it is difficult to assess whether the population we studied has reduced the amount of self-reported PA to a smaller or larger extent. The only available data we found about self-reported PA during confinement comes from activity trackers, where it has been shown that in Europe the country with the greatest step count decrease recorded was Spain, with 38% less, followed by Italy with 25%. This reduction is similar to the falling trend in walking time shown in our study. In this regard, other studies carried out in similar lockdown situations (confinement during seasons with adverse weather conditions—cold winter or heat waves) also reported PA decrease during confinement, whereas sedentary time increased because poor or extreme weather becomes an environmental barrier to going outdoors [14,15,16,17].

Sedentary time increased considerably, most likely due to the exchange between common daily active behavior (walking, cycling or transport to work, etc.) and the prolonged stay at home. Young people and students spent more time seated during confinement and this may be due to the forced e-learning which encourages sedentary behavior related to excessive time on screen-based activities [18]. Likewise, according to the socioecological model [11], in a comparable framework where social or environmental barriers promote an inactive lifestyle (social isolation, loneliness and, when season changes), more sedentary behavior and less time being spent on light, moderate and vigorous PA have been reported [19,20,21]. Hence, an involuntary prolonged stay at home may encourage sedentary behavior as well as during confinement caused by COVID-19.

The most active subjects showed the highest decrease in vigorous activity time and this may be explained by two main reasons: the forced sudden inaccessibility to community resources (e.g., sports facilities, urban trails, parks, green spaces, etc.) and the lack of time to react during the first weeks of confinement to gather fitness resources, in order to continue engaging in regular activities at home.

Regarding moderate activities, our data showed that it barely changed in all the population and this could be attributed to the fact that some people may have been maintaining the minimum recommended time doing alternative activities at home. In this regard, the less active population increased vigorous and moderate activities during confinement, which could be related to the promotion of such activities by health institutions, fitness centers, the Internet and television by posting daily online workout routines. To our knowledge, there are no scientific articles supporting the benefits of promoting PA established during confinement. Some specific examples of such activities are Spanish national television (morning workouts), the free online classes offered by several institutions and fitness centers (using online platforms such as Zoom or Google Meet) and the recommendations made by health associations. Our results showed that there are people who found new ways of being more active during this lockdown, which could become best practices in the future, should a pandemic of similar characteristics hit again.

Active behavior changed according to gender. Both genders reduced walking time to a similar extent, although men reported a higher decrease in vigorous activities that may be related to the greater PA prevalence reported in women over the years. [22,23,24,25]. In terms of sedentary behavior, sitting time increased more highly in men who also significantly reduced moderate activities, while women increased these activities. This could be attributed to the gender gap by historically demonstrated female inequality in household and child care tasks, in which men have shown a low level of involvement in Spain [26,27,28].

Regarding meeting PA recommendations, the global age-standardized prevalence among the inactive population was 27.5% in 2016 [23]. According to the Eurobarometer (Eurobarometer, 2014), 33.6% of the Spanish adult population did not attain minimum levels of PA, and 36% spent most of the day sitting down (national health surveys in Spain). In our population, our inactive population before confinement is lower (25.3% corresponding to 75 min/week vigorous PA) than that of the Eurobarometer. That may be explained by the fact that the population recruited for the study was particularly more active as shown in the data for people undertaking >225 min/week of vigorous activities, and this could be attributed to the fact that the questionnaire was shared by institutions linked to sport, exercise and biomechanics. Therefore, it is to be expected that we recruited a population that is commonly more active than the general population.

A limitation of the current study is that we selected the IPAQ short version instead of the IPAQ-Long one because of concerns that the length of the questionnaire would result in significant participant burden. In addition, although the questionnaire collects data about the last seven days, in our study we requested information beyond a week, which may be justified due to the unprecedented situation and the lack of time available to manage more appropriate study design and methods. In order to obtain as many participants as possible, sharing the questionnaire was sent to customers involved in sports and biomechanical issues in addition to being shared on social networks. Therefore, a sampling selection bias may have occurred in the population analyzed due to their natural active habits. Another limitation could be the cross-sectional design of the study since directionality of the associations cannot be established. Finally, the way in which the data were collected in this study can be in itself considered a limitation, since participants reported data in a subjective way when answering self-rated questionnaires.

On the other hand, one of the strong points of the study is that the questionnaire was sent just when the population started confinement (during the first two weeks), and so it is likely participants had their common activities in mind before confinement in a fresh and realistic way. To our knowledge, this is the first study carried out on a confined large sample size, which may be useful in designing different strategies in order for the confined population to reduce sedentarism and increase PA.

## 5. Conclusions

In conclusion, healthy adults reduced daily self-reported PA and increased sedentary time during COVID-19 confinement in Spain. The impact was particularly high in men, young people, students and the very active population. Strategies should thus be employed to increase PA and decrease sedentary behavior.

In this study, we tried to ascertain the influence of confinement on the lifestyle of the population. This study may serve to show the effect of an extraordinary measure, such as confinement, on this lifestyle, as well as a greater awareness of the effect of confinement. Moreover, the results may help design and target interventions aimed at increasing physical activity and reducing unhealthy levels of sedentary time associated with Pandemic mitigation efforts.

## Figures and Tables

**Table 1 ijerph-17-06878-t001:** Demographic characteristics of 3800 healthy subjects before confinement.

	*n* = 3800
Before Confinement	
Age (years)	42.7 ± 10.4
Height (cm)	171 ± 8
Weight (kg)	71 ± 12
BMI (kg/m^2^)	24.2 ± 3.2
Male/Female	2054 (54)/1746 (46)
Workers	2956 (78)
Students	267 (7)
Study–work	374 (10)
Nothing	203(5)
Data presented as mean ± SD, *n* (%)	

**Table 2 ijerph-17-06878-t002:** Self-reported physical activity (PA) data before and during confinement in all the population and different subgroups.

Subgroup Analysis	Before Confinement	During Confinement	*p* Value	Effect Size	Before Confinement	During Confinement	*p* Value	Effect Size	Before Confinement	During Confinement	*p* Value	Effect Size	Before Confinement	During Confinement	*p* Value	Effect Size
	Time in Vigorous Activities				Time in Moderate Activities				Walking Time				Sitting Time *			
Total (*n* = 3800)	219 ± 196	182 ± 184	<0.001	0.195 ^a^	149 ± 174	145 ± 170	0.102	0.023 ^a^	282 ± 253	116 ± 189.3	<0.001	0.743 ^c^	(*n* = 3687) 6.1 ± 3.6	8 ± 5.1	<0.001	0.430 ^b^
Women (*n* = 1746)	175 ± 176	159 ± 174	<0.001	0.091 ^a^	133 ± 160	144 ± 159	<0.05	0.069 ^a^	302 ± 260	122 ± 199.3	<0.001	0.777 ^c^	(*n* = 1694) 6.3 ± 3.9	7.9 ± 3.9	<0.001	0.410 ^b^
Men (*n* = 2054)	256 ± 204	202 ± 190	<0.001	0.274 ^b^	163 ± 185	145 ± 179	<0.001	0.099 ^a^	265 ± 247	110 ± 180.1	<0.001	0.717 ^c^	(*n* = 1993) 6 ± 3.1	8.1 ± 5.9	<0.001	0.446 ^b^
Workers (*n* = 2956)	212.1 ± 189.9	177.3 ± 179.4	<0.001	0.188 ^a^	143 ± 169.2	142.2 ± 170.6	0.811	0.005 ^a^	269.3 ± 246.2	113.7 ± 182.7	<0.001	0.718 ^c^	(*n* = 2865) 6.2 ± 3.5	8.0 ± 5.4	<0.001	0.396 ^b^
Students (*n* = 267)	295.5 ± 221.0	223.7 ± 199.1	<0.001	0.341 ^b^	171.1 ± 191.8	143.5 ± 157.1	<0.05	0.157 ^a^	298.8 ± 246.1	98.8 ± 189.7	<0.001	0.910 ^c^	(*n* = 262) 6.4 ± 2.4	8.8 ± 3.2	<0.001	0.849 ^c^
Study–work (*n* = 374)	223.6 ± 196.8	193.2 ± 195.2	<0.001	0.155 ^a^	157.4 ± 177.1	144 ± 160.6	0.141	0.079 ^a^	301.3 ± 249.5	106.1 ± 179.6	<0.001	0.898 ^c^	(*n* = 361) 6.3 ± 4.1	8.3 ± 3.4	<0.001	0.531 ^c^
Nothing (*n* = 203)	213.9 ± 228.4	179.6 ± 201.1	0.013	0.159 ^a^	198 ± 208.8	184.6 ± 190.4	0.316	0.067 ^a^	403.3 ± 326.3	186.6 ± 267.2	<0.001	0.727 ^c^	(*n* = 199) 4.4 ± 2.4	6.5 ± 3.5	<0.001	0.700 ^c^
Age categories																
18–24 (*n* = 264)	300 ± 206.6	246 ± 189.1	<0.001	0.273 ^b^	180 ± 197.3	149 ± 154.6	<0.05	0.175 ^a^	321 ± 281.8	94 ± 182.6	<0.001	0.956 ^c^	(*n* = 258) 6.6 ± 4.2	9 ± 3.5	<0.001	0.621 ^c^
25–34 (*n* = 670)	244 ± 197.9	201 ± 193.6	<0.001	0.220 ^b^	139 ± 150.3	145 ± 159.4	0.345	0.039 ^a^	280 ± 244.2	97 ± 161.1	<0.001	0.885 ^c^	(*n* = 655) 6.4 ± 3.1	8.6 ± 3.6	<0.001	0.655 ^c^
35–44 (*n* = 1361)	209 ± 189.9	175 ± 174.5	<0.001	0.186 ^a^	141 ± 176.6	140 ± 173.5	0.830	0.006 ^a^	253 ± 235.8	108 ± 186.8	<0.001	0.682 ^c^	(*n* = 1323) 6 ± 3.9	7.7 ± 3.9	<0.001	0.436 ^b^
45–54 (*n* = 1158)	202 ± 184.4	171 ± 183.3	<0.001	0.169 ^a^	150 ± 172.7	142 ± 170.7	0.121	0.047 ^a^	285 ± 256.1	125 ± 197.8	<0.001	0.699 ^b^	(*n* = 1113) 6.1 ± 3.1	7.9 ± 7.2	<0.001	0.325 ^a^
55–65 (*n* = 347)	199 ± 126	155 ± 186.1	<0.001	0.277 ^b^	169 ± 191.7	162 ± 184.3	0.405	0.011 ^a^	354 ± 284.1	160 ± 213.2	<0.001	0.772 ^c^	(*n* = 338) 5.7 ± 3	7.5 ± 3.5	<0.001	0.552 ^b^
PA categories for moderate activities																
0–150 (*n* = 2453)	187 ± 176.6	160 ± 168.3	<0.001	0.157 ^a^	49 ± 50.5	91 ± 119.6	<0.001	0.458 ^b^	254 ± 239.6	102 ± 180	<0.001	0.717 ^c^	(*n* = 2393)6.4 ± 3.6	8.2 ± 4.3	<0.001	0.454 ^b^
150–300(*n* = 838)	234 ± 178.8	196 ± 182.3	<0.001	0.210 ^b^	225 ± 45.2	197 ± 156.4	<0.001	0.243 ^b^	309 ± 252.3	121 ± 181.9	<0.001	0.855 ^c^	(*n* = 808) 5.7 ± 3	8.1 ± 7.3	<0.001	0.430 ^b^
300–450(*n* = 274)	299 ± 220	243 ± 210.8	<0.001	0.260 ^b^	386 ± 39	278 ± 204.7	0.709	0.733 ^c^	340 ± 272.8	167 ± 234.5	<0.001	0.680 ^c^	(*n* = 266) 5.7 ± 4.4	7.6 ± 3.3	<0.001	0.489 ^b^
>450 (*n* = 236)	409 ± 267.1	292 ± 243.5	<0.001	0.458 ^b^	643 ± 146.2	370 ± 266.1	0.010	1.272 ^c^	407 ± 308	179 ± 227	<0.001	0.843 ^c^	(*n* = 220) 4.8 ± 2.9	6.5 ± 3.3	<0.001	0.547 ^c^
PA categories for vigorous activities																
0–75 (*n* = 962)	16 ± 23.7	71 ± 124.7	<0.001	0.613 ^c^	101 ± 138.1	127 ± 153.9	<0.001	0.178 ^a^	291 ± 269.1	127 ± 211.8	<0.001	0.677 ^c^	(*n* = 937) 6.4 ± 3.2	7.9 ± 3.5	<0.001	0.447 ^b^
75–150 (*n* = 711)	115 ± 21.6	125 ± 116.2	<0.05	0.120 ^a^	119 ± 120.9	121 ± 132.8	0.596	0.016 ^a^	248 ± 219.5	93 ± 155.7	<0.001	0.815 ^c^	(*n* = 698) 6.5 ± 4.3	8 ± 3.5	<0.001	0.383 ^b^
150–225 (*n* = 550)	188 ± 16.4	172 ± 134.2	<0.01	0.167 ^a^	136 ± 146	128 ± 156.6	0.231	0.053 ^a^	256 ± 226.2	93 ± 153.8	<0.001	0.843 ^c^	(*n* = 528) 6.2 ± 3.1	8.2 ± 4.1	<0.001	0.550 ^c^
>225 (*n* = 1577)	400 ± 170.7	278 ± 203	<0.001	0.651 ^c^	196 ± 208	171 ± 193.7	<0.001	0.124 ^a^	299 ± 264.1	126 ± 197.5	<0.001	0.742 ^c^	(*n* = 1524)5.7 ± 3.3	8 ± 6.5	<0.001	0.446 ^b^

Data presented as *n*, mean ± SD. * Some variables pertaining to sedentary time have missing random values, as follows: total (*n* = 113); women (*n* = 52); men (*n* = 61); workers (*n* = 91); students (*n* = 5); study–work (*n* = 13); nothing (*n* = 4); age categories 18–24 (*n* = 6); 25–34 (*n* = 15); 35–44 (*n* = 38); 45–54 (*n* = 45); 55–65 (*n* = 9); Self-reported PA categories moderate 0–150 (*n* = 60); 150–300 (*n* = 30); 300–450 (*n* = 8); >450 (*n* = 16), PA categories vigorous 0–75 (*n* = 25); 75–150 (*n* = 13); 150–225 (*n* = 22); >225 (*n* = 53). Effect size calculated as the Cohen’s *d* categorized into ^a^ small size effect (<0.20) ^b^ medium size effect (0.20–0.50) and ^c^ high size effect (>0.50) for this paired *t*-test analysis.

**Table 3 ijerph-17-06878-t003:** Subgroup analysis in different ranges of physical activity levels before and during COVID-19.

Subgroup Categories (Time; Min/Week)	Before Confinement	During Confinement	Chi-Squared *p* Value
**Vigorous Activity**			
0–75	962 (25.3%)	1369 (36%)	<0.001
75–150	711 (18.7%)	650 (17.1%)	0.068
150–225	550 (14.5%)	495 (13%)	0.066
>225	1577 (41.5%)	1286 (33.8%)	<0.001
**Moderate Activity**			
0–150	2453 (64.6%)	2509 (66%)	0.117
150–300	837 (22%)	776 (20.4%)	0.087
300–450	274 (7.2%)	329 (8.7%)	<0.05
>450	236 (6%)	186 (4.9%)	<0.05
Data presented as *n* (%)			

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
