# Peer review of "Physical Activity Change during COVID-19 Confinement"

_ijerph, 2020, doi:10.3390/ijerph17186878_

Round 1

Reviewer 1 Report

Generally, this is an interesting and relevant study for the current time. However, the entire article is rushed. There is insufficient introduction, the methods were not sufficiently described, and data analyses carried out is too basic considering the wealth of data that the authors have.

Introduction:

One sentence should not form one full paragraph (first and last paragraph of introduction)

Introduction is rushed. The authors should still provide sufficient background information on physical activity and sedentary behavior. The authors should also provide theoretical reasoning for why the secondary outcomes were measured.

Methods:

I recommend that the authors section the Materials and Methods into design-setting-participants-materials-procedure-analysis.

Important: After reading the limitations in the discussion, only do we see that the questionnaire was modified, and asked for participants to report activity on two weeks. This is important to report here. The way it is reported now is extremely unclear.

Please state the exact numbers for excluded participants per criterion.

Why were the subgroups categorized as so (especially for the moderate and vigorous activity subgroups)? The authors report having eight groups for physical activity in the methods section, but only report on four groups in the results section. Which four groups are reported in the results section? What happened to the other four groups? It seems like the authors aggregated the eight groups into four, and if so, the authors should clarify how this was done.

The analyses are too basic. The authors have information of baseline activity and demographic information – these should be appropriately controlled for in the statistical models. Any data that are not missing at random should be controlled for as well.

Results:

For each subgroup, where are the results for the chi-square analyses? Did the subgroup differ from each other? Otherwise, why carry out a subgroup analysis? Are men’s physical activity and sedentary behavior significantly different from each other for example?

Multiple testing was done – did the authors correct for the risk of type 1 error?

Please report appropriate and comparable effect sizes for each analysis.

Table 2 could use a horizontal layout. The current layout is not reader friendly.

Did the authors check whether missing data was missing at random (MAR)? If MAR assumption is not satisfied, the authors should correct for this.

Discussion:

Acknowledge subjective limitation of self-rated questionnaires

Acknowledge sampling bias (tech savvy participants, and participants who were affiliated with the sports and biomechanical institutions).

The authors report in the limitation that data was collected beyond a week. This was not reported in the methods section and comes as a surprise. The assumption is that this means that the authors only carried out testing once, and asked the participants to recall their activities from two separate weeks? This must be made clearer from the onset, and include in the discussion the impact of such a modification.

Author Response

Point-by-Point Response to Reviewer’s Comments

We would like to sincerely thank the reviewers for their helpful recommendations. We have seriously considered all the comments and carefully revised the manuscript accordingly. Revisions are highlighted in yellow through the manuscript to indicate where changes have taken place. We feel that the quality of the manuscript has been significantly improved with these modifications and improvements based on the reviewers’ suggestions and comments. We hope our revision will lead to an acceptance of our manuscript for publication in International Journal of Environmental Research and Public Health.

In advance,

King regards

REVIEWER 1

General comments

Generally, this is an interesting and relevant study for the current time. However, the entire article is rushed. There is insufficient introduction, the methods were not sufficiently described, and data analyses carried out is too basic considering the wealth of data that the authors have.

Introduction:

REVIEWER: One sentence should not form one full paragraph (first and last paragraph of introduction)

AUTHORS: Thank you very much for your contribution. We have modified the first two paragraphs according to the update information we have nowadays. On the other hand, we have also modified the last paragraph of the manuscript in order to improve the structure of the article.

REVIEWER: Introduction is rushed. The authors should still provide sufficient background information on physical activity and sedentary behaviour. The authors should also provide theoretical reasoning for why the secondary outcomes were measured.

AUTHORS: Many thanks for the constructive suggestions on our manuscript, we have modified the last three paragraphs of the introduction following your suggestion.

Methods:

REVIEWER: I recommend that the authors section the Materials and Methods into design-setting-participants-materials-procedure-analysis.

AUTHORS: Thank you very much for your contribution. We have added the recommended sections.

REVIEWER: Important: After reading the limitations in the discussion, only do we see that the questionnaire was modified, and asked for participants to report activity on two weeks. This is important to report here. The way it is reported now is extremely unclear.

AUTHORS: Thank you very much for your contribution. In order to clarify and explain more clearly the way in which the data were collected we have modified the first paragraph of the experimental trials.

                “Sociodemographic data (age, height, weight, sex, whether working and/or studying) and self-reported physical activity data were collected by questionnaires sent  between 23rd March and 1st April, just 10 days after the state of emergency was declared in Spain. Two questionnaires were sent at one time to be answered consecutively at the same moment, the first collected retrospective data about habits during a normal week before the lockdown, and the second asked about the following one or two weeks.”

REVIEWER: Please state the exact numbers for excluded participants per criterion.

AUTHORS: Thanks for your suggestion. We have added the exact numbers of participants excluded per criterion. The new paragraph reads as:

“4160 healthy subjects answered the questionnaire, although 360 were not included due to exclusion criteria: not resident in Spain (n=16), age ≥ 65 or <18 (n=110), and extreme scores in vigorous, moderate and walking activities (≥6 hours per day 7 days a week). In the later reason, the exact number of participants excluded were as follows: n=81 were excluded because the sum of the total active time was >6h per day (vigorous + moderate + walking). Within the pre-COVID data, n=17, n=31 and n=62 were excluded because they exercised > 180 minutes per session in vigorous intensity, >180 in moderate intensity and > 21h (3h per day, 7 days of week) walking, respectively. Finally, within the during-COVID data, n=17, n=9 and n=17 were excluded because they exercised > 180 minutes per session in vigorous intensity, >180 in moderate intensity and > 21h (3h per day, 7 days of week) walking, respectively.”

REVIEWER: Why were the subgroups categorized as so (especially for the moderate and vigorous activity subgroups)? The authors report having eight groups for physical activity in the methods section, but only report on four groups in the results section. Which four groups are reported in the results section? What happened to the other four groups? It seems like the authors aggregated the eight groups into four, and if so, the authors should clarify how this was done.

AUTHORS: Thanks for catching up our error of missing data in the table. We apologize for such huge mistake. We have added in table 2 the missing data for the moderate intensity activity subgroups. The reasoning behind the subgroups categories comes from the World Health Organization (WHO) recommendations. We divided the data into different groups following the WHO recommendations for moderate (150 min per week) and vigorous (75 min per week) intensity activity. Under each of the two categories we divided the data into  4 groups which are: 1under recommendations (less than 150 min of moderate intensity or less than 75 min of vigorous intensity), following the recommendations (150 to 300 moderate intensity or 75 to 150 vigorous), two times the amount recommended, and three times the amount recommended.

REVIEWER: The analyses are too basic. The authors have information of baseline activity and demographic information – these should be appropriately controlled for in the statistical models. Any data that are not missing at random should be controlled for as well.

AUTHORS: Thanks for the suggestion. We certainly have data of baseline activity and demographic information. At first, we thought of different ways of analysing the data, including ANOVA, linear regression to predict activity behaviours and other analyses. However, our hypotheses were basic, and we wanted to report changes in time and % population prior and during confinement. For that reason, the way we approached the data was to divide the data into the subgroups (age, gender, working status, time spent in moderate or vigorous intensity activities at baseline). Having divided the groups into categories, the comparisons were basic as the reviewer points out. Nevertheless, the message that the data gives has a paramount importance and it is clearly reported in our modest opinion.

Results:

REVIEWER: For each subgroup, where are the results for the chi-square analyses? Did the subgroup differ from each other? Otherwise, why carry out a subgroup analysis? Are men’s physical activity and sedentary behaviour significantly different from each other for example?

AUTHORS: Thanks for the comment. The results for the chi-square are in table 3, we compared the change in % participants in each subgroup category for moderate or vigorous activity. Basically in table 2, we analyzed the data (paired t-test) to compare the time spent prior and during confinement in the different subgroups explained in the table, with the hypothesis that participants (by age, gender, working status or PA intensity) exercised less time during confinement. Table 3 shows the change in % of participants in those specific subgroups category because we wanted to explore deeper into that with our hypothesis that less people reach the WHO recommendations during the confinement.

REVIEWER: Multiple testing was done – did the authors correct for the risk of type 1 error?

AUTHORS: Thanks for your question. Despite the multiple comparison within the same participants we established an α of 0.05 which should be enough to correct for type 1 error. Nevertheless, we reported all the p-values in the table and most of them are very significant (<0.001) which give us reasons to believe that type 1 error was not incurred in the analysis.

REVIEWER: Please report appropriate and comparable effect sizes for each analysis.

AUTHORS: Thanks for the comment. We added the effect sizes as the Cohen´s d for each analysis of the table 2.

REVIEWER: Table 2 could use a horizontal layout. The current layout is not reader friendly.

AUTHORS: Thank you very much for your contribution. The document was delivered with the tables in a horizontal position. We do not know if your format can be affected in the delivery. We have re-positioned the table in horizontal, we hope that the formatting can be maintained.

REVIEWER: Did the authors check whether missing data was missing at random (MAR)? If MAR assumption is not satisfied, the authors should correct for this.

AUTHORS: Thanks for the comment. All the missing data were at random. We have reported it on table 2.

Discussion:                 

REVIEWER: Acknowledge subjective limitation of self-rated questionnaires

AUTHORS: Thanks for the positive and encouraging comments. We have added a sentence in the limitations.

Finally, the way in which the data were collected in this study can be a limitation, since participants reported data in a subjective way answering self-rated questionnaires.

REVIEWER: Acknowledge sampling bias (tech savvy participants, and participants who were affiliated with the sports and biomechanical institutions).

AUTHORS: Thanks for pointing out this issue. It is true that this way of diffusing the questionnaire can produce a sampling bias, and this can be seen, above all in terms of the registers of vigorous physical activity, that we are dealing with a very active population (https://www.mscbs.gob.es/estadEstudios/estadisticas/encuestaNacional/encuestaNac2017/ACTIVIDAD_FISICA.pdf), so we have added this fact as a limitation.

We have added a sentence in the limitations.  In order to obtain as many participants as possible, in addition to share the questionnaire by social networks, it was sent to customers related with sports and biomechanical issues. Therefore, a sampling selection bias may occur in the population analysed due to their natural active habits.

REVIEWER: The authors report in the limitation that data was collected beyond a week. This was not reported in the methods section and comes as a surprise. The assumption is that this means that the authors only carried out testing once, and asked the participants to recall their activities from two separate weeks? This must be made clearer from the onset, and include in the discussion the impact of such a modification.

AUTHORS: Thanks for the positive and encouraging comments. We have modified the section of experimental trials clarifying the way in which the data were obtained. Being aware of the limitation of collecting the data in this way, this fact is included in the limitations, although we believe that it may be justified by the situation we were in.

Reviewer 2 Report

Dear Authors,

Thank you for the interesting manuscript.

Below are my comments and suggestions mostly for the methods section.

You state that you collected data on self-reported weight and height. It might be interesting to add some information about average BMI and % of under-, normal-, and overweight/obese subjects and to compare these percentages to national surveys. This information could strengthen sample representativeness.

Did you have missing data? If yes, how did you treat them?

Please explain, when, and how written contents were collected from the participants?

And the most important question, how did you ensure that all the study participants filled in the questionnaire both times? Did you have some people lost after the first survey? This information should be added to the study organization description.

Please separate the statistical analysis description into a separate subsection.

Line 118: should it be 47.7%?

Please add practical implications based on your study results.

Author Response

Point-by-Point Response to Reviewer’s Comments

We would like to sincerely thank the reviewers for their helpful recommendations. We have seriously considered all the comments and carefully revised the manuscript accordingly. Revisions are highlighted in yellow through the manuscript to indicate where changes have taken place. We feel that the quality of the manuscript has been significantly improved with these modifications and improvements based on the reviewers’ suggestions and comments. We hope our revision will lead to an acceptance of our manuscript for publication in International Journal of Environmental Research and Public Health.

In advance,

King regards

REVIEWER 2

Dear Authors,

Thank you for the interesting manuscript.

Below are my comments and suggestions mostly for the methods section.

REVIEWER: You state that you collected data on self-reported weight and height. It might be interesting to add some information about average BMI and % of under-, normal-, and overweight/obese subjects and to compare these percentages to national surveys. This information could strengthen sample representativeness.

AUTHORS: Thanks for your recommendation. We added the BMI in the TABLE 1 of the manuscript. We have introduced it because of the importance it has to know the sample better. We added under table 1 the subdivision of the sample by BMI groups:

Using the WHO stratification, we divided the sample in 5 groups of BMI with n=77 underweight group, n= 2621 normal; n=897 over weight; n=144 obese and n=4 extremely  obese (<18,5; 18,5-24,9; 25-29,9; 30-39,9 and ≥40 groups of BMI score respectively)

REVIEWER: Did you have missing data? If yes, how did you treat them?

AUTHORS: Thanks for your question. All the missing data was at random, with such a big sample size and small number of missing data, we just reported in table 2 that all missing data was at random and pointed out the exact number of missing data.

REVIEWER: Please explain, when, and how written contents were collected from the participants?

AUTHORS: Thank you for your recommendation. The written contents of the participants were collected through a questionnaire sent in the second week of the confinement, which asks retrospectively about the habits before the confinement, and about the habits of the participants during the confinement that they were suffering. We have modified the first paragraph of the experimental trials with the aim of explaining more clearly how the data were obtained.

Sociodemographic data (age, height, weight, sex, whether working and/or studying) and self-reported physical activity data were collected by questionnaires sent  between 23rd March and 1st April, just 10 days after the state of emergency was declared in Spain. Two questionnaires were sent at one time to be answered consecutively at the same moment, the first collected retrospective data about habits during a normal week before the lockdown, and the second asked about the following one or two weeks.

REVIEWER: And the most important question, how did you ensure that all the study participants filled in the questionnaire both times? Did you have some people lost after the first survey? This information should be added to the study organization description.

AUTHORS: The questionnaire was sent out only once, asking twice about the participants' habits. The first time they were asked retrospectively about the habits they had before the confinement and the second time about the habits they had during the confinement they were suffering. We hope that with the sentence we have included in the text it has been clarified how we obtained the data.

REVIEWER: Please separate the statistical analysis description into a separate subsection.

AUTHORS: Thank you for your recommendation. We have separated the statistical analysis description into a separate subsection.

REVIEWER: Line 118: should it be 47.7%?

AUTHORS: Thank you for your question. We have added the % symbol. Line 127.

REVIEWER: Please add practical implications based on your study results.

AUTHORS: Thank you for your recommendation. We have added practical implications at the end reading as follows:

In this study, we tried to find out the influence of confinement on the lifestyle of the population. This study may serve to show the effect of an extraordinary measure, such as confinement, on the lifestyle of the population, as well as a greater awareness of the effect of confinement. Taking into account these results and their possible implications or effects on the health of the population, it can help to design plans with the aim of ensuring healthy living habits in similar situations

Reviewer 3 Report

A much needed and interesting study.
Please answer a few questions:
Was the survey addressed to the same people twice? Did the respondents evaluate only retrospectively - as before and during COVID-19 confinement? For the sake of clarity, please clarify if they were the same people twice.
On what basis was the size of the studied group estimated?
How is the representativeness of the test group ensured?

Yours sincerely

Author Response

Point-by-Point Response to Reviewer’s Comments

We would like to sincerely thank the reviewers for their helpful recommendations. We have seriously considered all the comments and carefully revised the manuscript accordingly. Revisions are highlighted in yellow through the manuscript to indicate where changes have taken place. We feel that the quality of the manuscript has been significantly improved with these modifications and improvements based on the reviewers’ suggestions and comments. We hope our revision will lead to an acceptance of our manuscript for publication in International Journal of Environmental Research and Public Health.

In advance,

King regards

REVIEWER 3

A much needed and interesting study.

Please answer a few questions:

REVIEWER: Was the survey addressed to the same people twice? Did the respondents evaluate only retrospectively - as before and during COVID-19 confinement? For the sake of clarity, please clarify if they were the same people twice.

AUTHORS: Thank you for your recommendation. The survey was sent to the same people twice, asking retrospectively about their habits before the confinement and asking about their habits during the confinement. We have modified the first paragraph of the experimental trials with the aim of explaining more clearly how the data were obtained.

Sociodemographic data (age, height, weight, sex, whether working and/or studying) and self-reported physical activity data were collected by questionnaires sent  between 23rd March and 1st April, just 10 days after the state of emergency was declared in Spain. Two questionnaires were sent at one time to be answered consecutively at the same moment, the first collected retrospective data about habits during a normal week before the lockdown, and the second asked about the following one or two weeks.

REVIEWER: On what basis was the size of the studied group estimated?

AUTHORS: Thanks for your question. Due to the logistics of the study we did not calculate a sample size or did a power analysis of the study. The way we collected the data is explained in the paper and we did not know how many people would respond to the survey. We were delighted to find out the final number of participants.

REVIEWER: How is the representativeness of the test group ensured?

AUTHORS: Thanks for your question. You mention a very important point which we have tried to clarify in the limitations of the study. There is a possible sampling bias which may imply that the representativeness of the group was not ensured. Here again, given the circumstances we think the data collected is worth publishing despite the possible limitations.

Round 2

Reviewer 1 Report

Overall, very interesting and important paper. Kudos to the authors for your upcoming contribution to society. There are just some final changes that the authors should address in my opinion:

Page 1 line 32 – Please introduce the severe acute respiratory syndrome coronavirus 2 before using the abbreviation SARS-CoV-2, and the coronavirus disease 2019 before using COVID-19. Although it may be common knowledge now, your study may be read by another generation many decades (and many pandemics) from now, so it is better to be clear and precise.

Page 2 line 44 – may be too picky here, but I don’t recommend using “significant concern” in a scientific paper unless this concern has been statistically tested

Page 2 line 47 – be careful with your wording. The WHO recommends 150 min moderate, 75 min vigorous, or “an equivalent combination” of both. Just “a combination of both” may mislead the layperson into thinking that the recommendation is 75+150 minutes.

Page 2 line 66 – why specifically 64 and below?

Page 5 – how did the authors check for missingness at random (MAR)?

Page 6 lines 158-163. The authors start by saying “To our knowledge, there are no previous studies that have analysed the effect of confinement caused by a pandemic on physical activity.” And two sentences later, the authors contradict their claim with “The only available data we found about physical activity during confinement comes from an activity tracker, where it has been shown that in Europe the country with the greatest step count decrease recorded was Spain, with 38% less, followed by Italy with 25%.”. This is merely the phrasing that the authors have used. Instead of saying that there are no studies out there, the authors should soften the claim, by saying that there are limited/very few/emerging studies etc. Also, the data that comes “from an activity tracker” is once again a shocking statement. The data came from multiple activity trackers. I think the authors meant “an activity tracking study”.

Page 7 lines 77-87 – the authors used a lot of speculation. Is there data to back this up? If there are no scientific articles regarding this, perhaps news sources, WHO recommendations, corporate company’s research and promotion and etc. For example, the authors write “could be related to the promotion of such activities by health institutions, fitness centres, the Internet and television by posting daily online workout routines.”. The authors could specify here, what type of promotion and by who exactly? Because these are best practices – the population did not drop in their moderate activities. Should a pandemic hit in the future, we would like to advise the future generation of what to do to maintain moderate activities.

Author Response

REVIEWER 1

General comments

Overall, very interesting and important paper. Kudos to the authors for your upcoming contribution to society. There are just some final changes that the authors should address in my opinion:

REVIEWER: Page 1 line 32 – Please introduce the severe acute respiratory syndrome coronavirus 2 before using the abbreviation SARS-CoV-2, and the coronavirus disease 2019 before using COVID-19. Although it may be common knowledge now, your study may be read by another generation many decades (and many pandemics) from now, so it is better to be clear and precise.

AUTHORS: Thank you for your contribution. We have modified and mark in yellow the text following your instructions.

On 11th March 2020, the World Health Organization (WHO) [1] declared a global pandemic caused by severe acute respiratory syndrome coronavirus (SARS-Covid-2), which has become a public health emergency of international concern. During its first phase of expansion outside China, Italy and Spain were the most affected countries reporting most cases and deaths. Thus, they were the first nations to declare a state of emergency in Europe. In Spain it was declared on 17th March and the government ordered a lockdown. to restrict travel and cancel non-essential services in order to stop the spread of coronavirus disease (COVID-19) [2].

REVIEWER: Page 2 line 44 – may be too picky here, but I don’t recommend using “significant concern” in a scientific paper unless this concern has been statistically tested.

AUTHORS: We agree with your recommendation. We have changed the word “significant” for “general”.

Likewise, there is a general significant concern about the negative health implications that inactivity and  sedentary behaviour may promote

REVIEWER: Page 2 line 47 – be careful with your wording. The WHO recommends 150 min moderate, 75 min vigorous, or “an equivalent combination” of both. Just “a combination of both” may mislead the layperson into thinking that the recommendation is 75+150 minutes.

AUTHORS: Thank you for this important detail. We have modified the sentence following your advice.

The general recommendation to consider an adult physically active is to achieve at least 150 minutes of moderate or 75 minutes of vigorous intensity activity per week, or an equivalent combination of both

REVIEWER: Page 2 line 66 – why specifically 64 and below?

AUTHORS: Thanks for your comment. We decided until 65 years because in the World Health Organization (WHO) physical activity recommendation are separated in 3 age range groups, from 5-17 years, 18-65, and older than 65. Even though the recommendations are very similar between the last two groups we decided to separate them and focus on the 18 to 65 to be more accurate on the results of the studied sample. We specified it on the manuscript: “Healthy adults (age ≥18 - ≤64 years, age category defined by WHO)”

REVIEWER: Page 5 – how did the authors check for missingness at random (MAR)?

AUTHORS: The authors knowledge on the field was the first mechanism to check for missing data at random. As the reviewer and readers may check, all the missing data were about sedentary time which could be a sensitive issue in many cases. However, it is also the last question of the survey which could be the reason to skip the answer and did not have any relation with the observed data from those individuals at a first look into the data. Nevertheless, the authors run chi-square tests between the variable and the other variables in the data to see if the missingness on the sedentary time was related to the values of the other variables. SPSS has a missing data module that helps in this matter.

REVIEWER: Page 6 lines 158-163. The authors start by saying “To our knowledge, there are no previous studies that have analysed the effect of confinement caused by a pandemic on physical activity.” And two sentences later, the authors contradict their claim with “The only available data we found about physical activity during confinement comes from an activity tracker, where it has been shown that in Europe the country with the greatest step count decrease recorded was Spain, with 38% less, followed by Italy with 25%.”. This is merely the phrasing that the authors have used. Instead of saying that there are no studies out there, the authors should soften the claim, by saying that there are limited/very few/emerging studies etc. Also, the data that comes “from an activity tracker” is once again a shocking statement. The data came from multiple activity trackers. I think the authors meant “an activity tracking study”.

AUTHORS: We agree with the reviewer. We have modified the both sentences to clarify the confusion reported by the reviewer.

To our knowledge, there are limited  no previous studies that have analysed the effect of confinement caused by a pandemic on physical activity….

The only available data we found about physical activity during confinement comes from an activity trackers.

REVIEWER: Page 7 lines 77-87 – the authors used a lot of speculation. Is there data to back this up? If there are no scientific articles regarding this, perhaps news sources, WHO recommendations, corporate company’s research and promotion and etc. For example, the authors write “could be related to the promotion of such activities by health institutions, fitness centres, the Internet and television by posting daily online workout routines.”. The authors could specify here, what type of promotion and by who exactly? Because these are best practices – the population did not drop in their moderate activities. Should a pandemic hit in the future, we would like to advise the future generation of what to do to maintain moderate activities.

AUTHORS: Thanks for the comment. We know that for example the Spanish public tv channel (TVE) offered every morning a workout with a physical exercise instructor. Also, many of the fitness centres adapted their workouts from “presential at the site” to online using platforms like Zoom of Google Meet. Finally, some health associations as endocrinology, nutrition and physical exercise group, recommended workouts to maintain the general population active. We have included some examples in the text:

To our knowledge, there are no scientific articles supporting the benefits of the promotion physical activities created by for example the Spanish national television (morning workouts) or the free online classes offered by several institutions and fitness centers (using online platforms like zoom or Google Meet) or the recommendations made by health associations, however our results showed that there are people that have found new ways of being more active during this lockdown. Those could be best practices in the future should a pandemic of similar characteristics hit again.”  

Reviewer 2 Report

Dear Authors,

Thank you for the improved version of your manuscript. I have a few more minor comments.

Table 1. Please provide a measurement unit of BMI. Please provide % of people in BMI subgroups.

Lines 108-109. As the WHO for health benefits recommends 75-150 min/week of vigorous or 150-300 min/week of moderate PA or equivalent combination of vigorous and moderate PA, it remains unclear based on what criteria these categories were created? Please add some explanation.

Author Response

REVIEWER 2

Dear Authors,

Thank you for the improved version of your manuscript. I have a few more minor comments.

REVIEWER: Table 1. Please provide a measurement unit of BMI. Please provide % of people in BMI subgroups.

AUTHORS: Thanks for the observation. We added the unit (kg/m2) and the percentages of BMI in the manuscript (table 1 and lines 143 to 145).

REVIEWER: Lines 108-109. As the WHO for health benefits recommends 75-150 min/week of vigorous or 150-300 min/week of moderate PA or equivalent combination of vigorous and moderate PA, it remains unclear based on what criteria these categories were created? Please add some explanation.

AUTHORS: Thanks for your comment. We had added some explanation to the categories in the text, line 109.

The reasoning behind the subgroups categories comes from the World Health Organization (WHO) recommendations. We divided the data into different groups following the WHO recommendations for moderate (150 min per week) and vigorous (75 min per week) intensity activity. Under each of the two categories we divided the data into  4 groups which are: 1) under recommendations (less than 150 min of moderate intensity or less than 75 min of vigorous intensity), 2) following the recommendations (150 to 300 moderate intensity or 75 to 150 vigorous), 3) two times the amount recommended, and 4) three times the amount recommended. This categorization was made to extrapolate the results to the amount of PA achieve by the participants of the study.
